# Mass Customisation for Zero-Energy Housing

Pablo Jimenez-Moreno 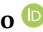

Low Carbon Building Group, Oxford Brookes University, Headington Campus, Oxford OX3 0BP, UK;
pjimenez-moreno@brookes.ac.uk

**Abstract:** This article describes the potential that co-design and marketing strategies have on increasing the consumption of energy-efficient dwellings. It explains how Japanese housebuilders are using 'mass customisation'—a phenomenon that mirrors the production and marketing of the automobile sector—in order to produce zero-energy houses and how this applies to the UK. The research consisted of a comparative analysis of Japanese and UK housebuilding. It identifies how mass customisation strategies are used to drive the sales of zero-energy houses in Japan and infers how to apply these in the UK context. This research found that some housebuilders in the UK are already using production strategies that resemble Japanese practices; however, the sustainable benefits observed in the Japanese context are not present in the UK because housebuilders' mass customisation strategies are limited to construction and not used as part of the marketing, co-design, and selling processes. Production and consumption of sustainable houses would increase in the UK if housebuilders implemented full mass customisation, meaning selecting existing robust production processes, defining an appropriate space solution, and using informative navigation tools.

**Keywords:** housing; mass-customisation; sustainability; zero-energy; prefabrication; user-oriented design; agile and lean manufacturing

## 1. Introduction

The current options for sustainable housing in the UK open market are extremely limited [1–4]. The present examples of sustainable practice consist of isolated ventures of individuals with the dedication and time to employ architects (or constructors) for what are essentially bespoke services [5–7]. The benefits of standardisation regarding quality, as defined as consistency, price certainty, and production efficiencies, are lost, causing most house-buyers to opt for mass housing options [8,9].

In contrast, Japanese manufacturers sell houses on-demand, allowing customisation in detail, including energy efficiency features [10,11]. The building energy costs and environmental impact are seamlessly communicated to the customers with brochures, models, and visual devices that allow them to make informed choices regarding house design and performance [12]. With such an approach comes many benefits rarely seen in UK housebuilding, e.g., high levels of energy efficiency and personalisation. Japanese house manufacturers are leading the production of zero-energy and zero-carbon houses [13–16].

Mass customisation is presented as a path towards a sustainable development in the UK housing context, which could cover not only environmental aspects but also increase user satisfaction and happy living.

This article aims to describe how mass customisation is related to the high production of zero energy houses in Japan. It also aims to explain how it could be applied to the UK by identifying suitable aspects of its housing practice, understanding the similarities and polarities of both contexts. It also describes one of the multiple connections that industrial production has with housing, sustainability, and improvement of user's life [17–22], joining a current movement that positions manufacturing as a key element for solving environmental and housing shortage issues in the UK [23–26]. This article proposes the use

of mass customisation design and marketing strategies as a solution to increase the production of sustainable houses without the need for increasing the UK's existing industrial manufacturing capacity.

## 2. Theoretical Background

### 2.1. Definition of Terms

2.1.1. Zero Energy

'Zero energy' is a conceptual term that refers to the balance of energies affecting a system [27–30]. In physics, energy refers to the capacity of a system to do work that can be referred to various settings [31]. The ambiguity of energy, as considered in physics, does not correspond to the context of operational energy of a dwelling that consists of standard electrical and gas measurements [32]; as Masa Noguchi [33] stressed that '...a kilowatt is a kilowatt...'.

A literal interpretation of zero energy as the absence of energy is a misunderstanding as buildings are defined by dynamism, and thus a point of true energy balance is highly elusive [31]. Consequently, zero energy is a concept of balance rather than a limit. A threshold rather than an exact equilibrium, where the total 'negative' energy is equal or higher than the 'positive' energy, considering carbon-free energies as negative. PlusEnergy buildings or energy-plus houses are exceptions where the values are inverted.

For the built environment, zero energy is translated as a building that produces as much energy as it consumes, also known as 'zero-energy buildings' (ZEBs) [34–39]. Zero-energy standards focus on operational energy, rather than on embodied energy, on the basis of the fact that 80% of the energy is consumed during the operational phase of a dwelling [40]. The factors determining the definition of a ZEB are:

1.  Energy balance—the energy balance over a fixed period of time [41].
2.  Grid connection—position of the buildings' energy connection to the grid [42,43]. The zero-energy balance implies a connection to an electrical grid; therefore, it really refers to 'net' energy. This article omits the word 'net' to avoid redundancy. Buildings not connected to the electricity grid, known as 'off-grid', offset all their energy consumption through their mediums [44].
3.  Metric—units used to measure energy content, with kWh being the most used in the operational domestic context [45];
4.  Balancing period—the period over which the energy balance is calculated or measured [28]. The energy calculations adjust to established time spans. Annual calculations are the most commonly used.
5.  Balance type—criteria used to verify the energy balance determined by the building boundary, energy generators, and energy consumers [46].
6.  Energy usage coverage—zero-energy standards usually omit gas in the energy equation as it is not possible to produce gas within the domestic realm to generate a balance and focus only on electricity.
7.  Generation type—the ways of generating energy. Carbon-free processes, or renewables balanced with carbon-based prime sources. Most energy standards contemplate all electricity imported from the grid as carbon-charged.
8.  Spatial boundary and generation location—the point where the building interacts with the electric grid usually matches where the meter locates.

Accordingly, a zero-energy house can be defined as an energy-efficient dwelling that generates enough electricity on-site over a year to supply all expected on-site energy services for the dwelling users [46]. In theory, a dwelling does not require to be energy-efficient to achieve the energy balance. However, zero energy is a concept rooted to environmental principles, and therefore it makes sense to merge these as one [34,47–49].

### 2.1.2. Mass Customisation

Mass customisation refers, from a production perspective, to the ability to provide customised products, or services, for individuals at scales, costs, or efficiencies that resemble 'mass production' [50–57].

On one hand, mass production refers to the single-purpose manufacture system that results in smooth flow of materials, large-volume production, and low prices, usually characterised by the use of linear manufacturing organised by a transport system or conveyor line [58,59]. On the other hand, customisation refers to the production of bespoke objects through crafted processes, in which times and sequences are not standardised or synchronised, making each production and product different [60,61].

Accordingly, mass customisation is considered a sophistication of mass production, just as mass production is considered an evolution of the crafting system [60,62–64]. Mass customisation is a paradigm used by companies to adjust their manufacturing processes to adapt to market subdivision due to increasing cultural and social heterogeneity to keep their production finances stable and healthy [65–68].

Mass customisation is presented as a solution for providing products and services that meet the needs of each customer concerning certain product features, where operations are performed within a fixed solution space, characterised by stable but still flexible and responsive processes to customisation that does not imply a switch in an upper market segment [69]. However, in practice, mass customisation is complicated to apply because it requires the involvement of the customer (end-user) during the production process, and therefore implies the postponement of production and manufacturing dependent to customers' orders [70–73]. The strategies used to overcome these challenges are known as mass customisation capabilities, which are defined as follows:

1.  Solution space—pre-existing capability and degrees of freedom built into a given manufacturer's production system framing the production extents of customisation. The solution space determines the universe of outcomes that a producer provides to their customers, and within that universe, specific product permutations are provided. Mass customisation does not mean to offer limitless choice but provides a choice restricted to options in the system's capacities [74].

2.  Robust process—the capability to reuse, re-arrange, or re-combine existing organisational and value-chain resources to fulfil a system of differentiated customers' needs [50,71,75]. A robust process can be achieved by having (1) flexible automation and modular processes which can be quickly and easily re-tasked depending on design change [76]; (2) adaptive human resources, where employees can deal with novel and ambiguous tasks to offset potential rigidities [77]; and (3) the supply chain separated into stages to postpone production and define product differentiation between fixed and flexible production stages. The point that separates decisions made under uncertainty (custom) from decisions made under certainty (standardised) is called 'customer order decoupling point' (CODP); its positioning in the supply chain sets the balance between productivity and flexibility [78–80]. The closer the CODP is from the supply perspective, the higher the customisability. Likewise, the closer to it is the demand perspective, the higher the production control, as shown in Figure 1 [55,81,82].

3.  Choice navigation—capabilities of a company to enable and support the customers to identify and customise their product by minimising the complexity and burden of choice. It refers to the interface where customers explore and decide on the producer offerings [83].

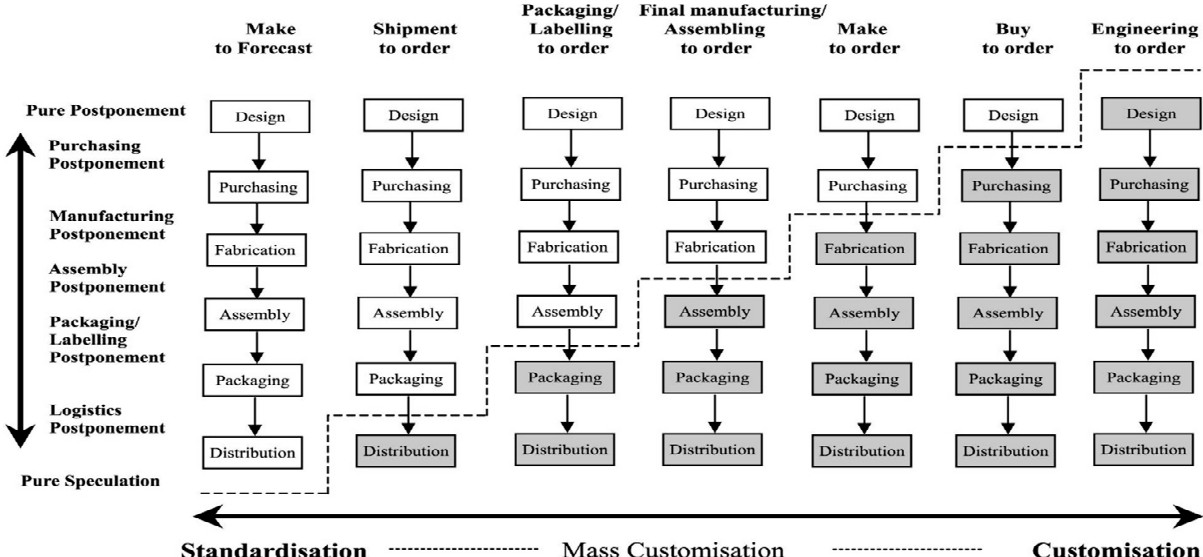

**Figure 1.** Customisation level determined by supply chain postponement (positioning of COPD).

Concretively, mass customisation refers to the co-design processes of products and services that allow end-users to customise their products to certain limits that perform within the mass customisation capabilities—solution space development, robust process, and choice navigation—to ensure stable but still flexible and responsive processes. The implementation of mass customisation requires the use of technologies or methodologies known as mass customisation 'enablers' [53,84,85].

### 2.1.3. Agile and Lean Manufacturing

The full application of mass customisation requires the implementation of agile and lean manufacturing systems [86–88]. Agile manufacturing relates to the principles of customisation, as it refers to the capability of surviving and prospering in a competitive environment of continuous and unpredictable change in markets by reacting quickly and effectively, driven by customer-designed products and services [89]. Lean manufacturing supports the development of mass customisation in reducing the impact of customer choice and productivity. It refers to the manufacturing methodology that focuses on maximising production value and productivity by minimising waste and inventory, quality control, and continuous improvement from staff, suppliers, and customers' feedback [90–92]. Lean manufacturing allows for the supplying of exactly the right quantity at the right time, and exactly the correct location in the production process, also known as the 'just-in-time' technique.

### 2.2. Mass Customisation Relation to Sustainable Housebuilding

Housebuilding today almost inevitably uses components and tools produced via industrialised processes. The production of energy efficiency components, such as photovoltaics, wind turbines, efficient heating/cooling systems, hermetic double/triple glazing windows, insulation panels, and mechanical systems, is dependant to manufacturing processes. Neglecting the mass production side of the sustainable building would be counterproductive [93]. Moreover, industrial manufacturing of construction components contributes to reduction of waste compared to its production on site.

On its part, sustainable design requires adaptation (custom) to the orientation and climatic and micro-climatic conditions of each site, as well as to the energy legislations of each context.

In brief, a main aspect of sustainable design consists of including energy-efficient components produced through mass production processes regarding to the particularities of each site and its users, and therefore needs to be custom as well as mass produced [6,7].

### 3. The Japanese and UK Contexts

Today, Japanese housing manufacturers are highly recognised for the use of mass customisation, as well as for leading the commercialisation of zero-energy and zero-carbon houses [94–101]. Interestingly, in Japan, the term mass customisation is not commonly used. It appears to be deeply woven into the Japanese organisational culture and service thinking, which might block its use in everyday language or as a topic of scientific concern [1,12,62,101].

It is important to understand which aspects of the Japanese housebuilding practice are exclusive to its context to visualise which aspects have potential application in the UK. Accordingly, it is essential to understand which aspects of the UK housebuilding practice are exclusive to its context to visualise the aspects not suitable for implementing Japanese mass customisation strategies [2,66,102].

*3.1. Historical Comparison of Housing from Postwar Times to the Present Day*

After the Second World War, Japan and the UK were in need of housing in order to recover from the urban destruction produced by the war. Both countries initially opted for the industrialised process to overcome their substantial housing deficits [103]. However, only Japan maintained a high percentage of its housing production through industrialised processes [104].

By 1945, Japan had a shortage of 4.2 million houses, where destruction accounted for over 30% of their urban environment [105]. Governmental efforts and policies focused on rebuilding the national economy and concentrated resources into strategically selected industries, which did not include housing [106,107]. In contrast, the UK had the industrial capacity and resources to set immediate housing programmes, allowing the construction of over 155,000 prefabricated bungalows, also known as 'prefabs' [108]. Between 1945 and 1951, local authorities using highly industrialised construction systems built 89% of the houses in the UK. However, in 1949, the prefab programmes were cancelled, and with it all manufacturers stopped producing houses and focused on other markets [109–111].

In Japan, with the rise of their economy after the 1950s, manufacturers invested in the production of prefabricated housing [112,113]. Initial houses were austere and lower in quality compared to the UK, justified with the high demand [114]. Eventually, the manufacturing housing industry consolidated; most house manufacturers active today were funded in the decades of the 1960s and 1970s [115,116]. In the early 1970s, housing production increased rapidly, ending the housing shortage carrying on from war times [106]. Different political and economic factors caused land prices to inflate and eventually collapse in the 1990s. Since then, Japanese house manufacturers have pursed mass customisation strategies to compete in a housing market dominated by self-build construction, where customer choice and quality are prioritised. Today, house manufacturers provide 15% of Japanese housing, mainly through self-build processes [117].

In the decades of the 1960s and 1970s in the UK, housing supply was mainly covered by the construction of housing estates. However, the subsidies to these activities stopped due to bad reputation caused by low quality and catastrophic events such as Ronan Point [118,119]. Since then, housing supply in the UK has relied on the private speculative sector. This restrictive housing model benefits from monopolising land and reducing construction costs causing low productivity [120], low satisfaction levels [23], and lack of investment in R&D and innovation [121,122].

*3.2. Land Effect on Housing Processes*

Japan has particular geologic and topographical conditions that in combination with its high population limit the availability for urban settlement to only 5% of the territory for urban settlement. Agricultural land (13%) is not commonly transformed into urban areas as Japan imports most of its food [123]. In the late 1980s and early 1990s, Japan experienced a radical estate price inflation driven by speculative behaviours. Residential land prices peaked in 1991 and then dropped from 1992 to 2005, being stable since then [124].

Dwellings and land are valued separately, causing dwellings to lose value with time. Japanese homeowners find it more economically attractive to change rebuild their houses in their land rather than relocating, favouring the self-build sector, which counts for 75% of the housing market. In a different manner to the UK, Japanese housebuilders cannot control housing demand, and therefore need to compete in quality, customer satisfaction ('green' market), performance, style, branding or marketing, resistance against disasters, and efficiency to cover customers' wants and needs [125,126].

The UK has a housing deficit of 250,000 dwellings per year, not as a matter of land availability as agricultural land covers over 75% of the territory. The land is restricted because housing developers hold stocks of land without developing them; today, there is a current stock of 500,000 unbuilt plots with planning permission [127,128]. Developers' financial success relies on their ability to buy land at low prices, maximise their value through speculation, and reduce construction costs [129,130]. In the UK, 90% of new houses are built through processes of land speculation, while in Japan, this number is only 25% [131]. In the UK, properties are valued as the entity of dwelling and land, and therefore speculative control provokes a constant rise in housing value. The average price of a house in the UK in 1971 was GBP 5362, less than 3% of today's average price, which is over GBP 200,000 [3,132,133].

### 3.3. Planning Systems

Japanese legislation protects landowners with very low restrictions and simple planning processes to build for residential purposes. The Japanese laissez-faire legislation is designed to encourage the fast reconstruction of housing stock to keep the housing industry as a productive business. Planning areas are categorised into 12 zones defined by nuisance. Single houses are considered to be low nuisance level, and thus are allowed in 11 of the 12 zones [134,135]. In addition, Japanese regulations demand a physical gap between buildings to protect them against earthquakes and fire, provoking the dominance of detached housing. Consequently, houses are built almost anywhere and in any form and typology, raising the need for customisation [136].

In the UK, the planning system provides control to local councils over landowners. Dwellings' typology, form, shape, style, and zoning must be consistent with legislation and sensible to the character of the surrounding existing stock. Innovative dwelling proposals suffer from long and complicated planning processes. The UK does not use zoning as a planning technique; permissions are given by the planning committees following internal protocols and on their own criteria [137–139]. Consequently, self-builders need to focus their services to ensure planning permission first and foremost.

### 3.4. Housing Need

Housing need refers to the characteristics of the housing needed by the population, including cost, location, and type. Japan overcame its housing deficit in the 1970s; still, the housing industry supplies over 950,000 houses every year (see Figure 2). Japan builds six times more houses than the UK per year, even with an ageing and declining population [140]. Japan's housing starts are high compared, not only to the UK but also to other countries. It has comparable housing starts to the USA with only 40% of the population [141].

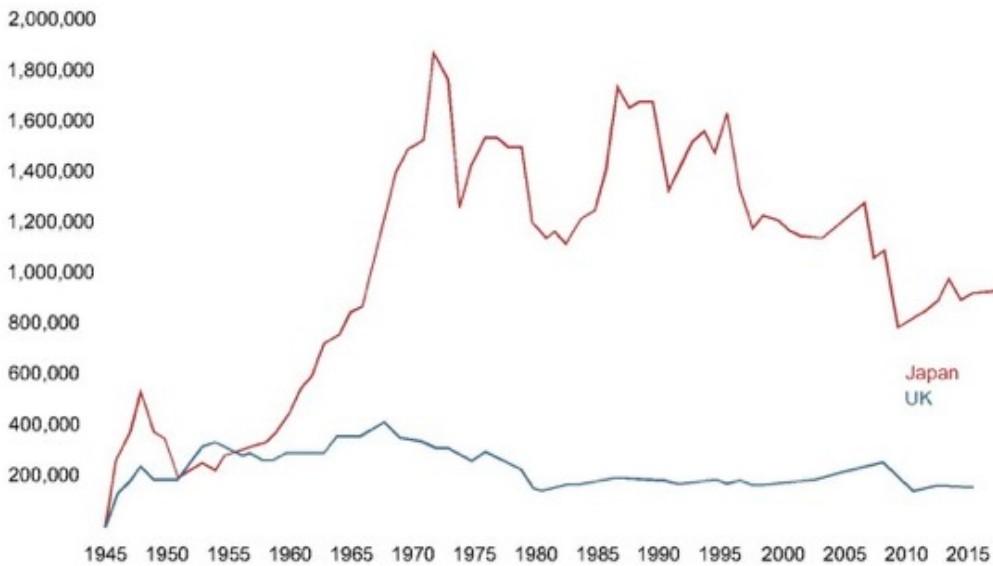

**Figure 2.** Housing starts in Japan and the UK from 1945 to 2015.

Japan requires a high production of houses due to the short lifespan of their buildings, constant increase of construction standards, and preference for new housing. The average age of houses in Japan is estimated to be 30 years, which is highly related to the constant exposure to natural disasters [119,142]. Another phenomenon that affects the longevity of houses is that they are valued separately from the land, encouraging landowners to renew (rebuild) the dwelling in their plots to bring value back to their properties [95,96,143,144]. In Japan, selling a second-hand house is rare; new-build houses count for around 80% of the total housing transactions.

In the UK, housing starts are low, not only in comparison to Japan but against other countries in Europe. It is estimated that the UK needs 400,000 new houses per year, but the housing industry only supplies 150,000 due to the dominance of speculative housing processes (see Figure 2). As a result, there is an annual deficit of 250,000 houses, which has increased by 50% in the last 10 years [145,146]. In the UK, properties constitute the entity of house and land, and are valued accordingly. Property prices keep rising. New-build dwellings are not perceived to provide advantages in comparison to existing stock, counting for only 5% of the total housing transactions. Rebuilding is an unnecessary expense as the value of properties mainly increases with land inflation rather than on the buildings' condition [128].

*3.5. Impact on Energy Efficiency*

The UK has opted for a 'fabric first' approach, having mandatory U-value standards for new dwellings. All new dwellings in the UK need to be under 0.18 $W/m^2K$ (walls). Other energy-related regulations have been dropped out because the construction industry has not been able to cope with them, forced by speculators to keep costs low; by 2015, the Green Deal was scrapped, and the Code for Sustainable Homes withdrawn [147,148].

In Japan, only buildings with areas over 300 square metres need to comply with energy regulations. New houses in Japan have an average area of 125 square metres. Only around 25% of all new housing starts are evaluated per year. Japan possesses attractive funding programmes and grants that promote higher energy performance of new buildings, causing 80% of them to meet environmental policies voluntarily [149–151]. Energy legislation focuses on regulating the energy used by appliances rather than fabric, including a mandatory minimum efficiency standard for all machinery, equipment, and appliances [152]. Incentives for domestic production are high to cope with energy crises that Japan has suffered in the last 40 years, such as the oil crisis of 1973 and shutdown of nuclear plants.

Japanese house manufacturers comply with energy standards as a marketing strategy; when the 'Housing Performance Indication System' and the feed-in-tariff legislation were implemented in 2000 and 2002, respectively, houses delivered with photovoltaics (PV) increased from 539 to 52,863 from 1994 to 2003 [14,100].

## 4. Materials and Methods

This research used a 'triangulation' method as the topics in concern—mass customisation, energy-efficiency, and housing—are different academic disciplines and the study of varied disciplines favours the use of different methodologies [153]. There is an increasing tendency in research to use 'mixed methods', not only for its effectiveness and practicality but as a process of increasing the scope of the research and avoid bias [154–160].

This research concretively analysed 6 companies, 3 in each context: Daiwa Homes, Sekisui House, and Sekisui Heim in Japan, and Robertson, Scotframe, and Carbon Dynamic in the UK, through fieldwork visits to the manufacturing facilities (Table 1). Daiwa Homes have policies for protecting their manufacturing processes, and therefore could not be included in this research. However, Daiwa Homes was still included in this research as it recognised for having the most user-oriented co-design system and for providing the most customised houses.

**Table 1.** Fieldwork sites and unit of study.

|  | Date | Company/Organisation | Location | Type of Facility |
|---|---|---|---|---|
| Japan | May 2015 | Sekisui House | Kanto, Koga, Ibaraki | Building prototypes<br>Manufacturing facilities<br>Housing park<br>Manufacturing facilities |
|  |  |  | Kizugawa, Kyoto | Information centre |
|  |  | Sekisui (Heim) Chemical | Toyohashi | Manufacturing facilities<br>Show home and selling centre |
|  |  | Daiwa House | Nara | Museum<br>Information centre<br>Research and development centre |
| UK | March 2016 | Carbon Dynamic | Invergordon, Scotland | Manufacturing facilities |
|  | March 2017 | Scotframe | Cumbernauld, Scotland | Manufacturing facilities<br>Design and engineering offices<br>Showroom |
|  | June 2017 | Robertson | Seaham, England | Manufacturing facilities |

In detail, data were collected through direct observations (qualitative and quantitative)—recordings (photo and video) and personal notes; documents and reports review (qualitative and quantitative), including documents produced by the organisations, such as brochures and reports to complemented data collected from the fieldwork; case studies (qualitative); and interviews (qualitative). Grounded theory was the method used to analyse data obtained from the literature review and fieldwork to elaborate on the research argument.

The data were coded using a three-phase coding system. First, an 'open coding' method was used, where data were collected using predesigned matrices [161]. Spreadsheet tables were printed and filled up in the fieldwork sites. Then, an 'axial coding' method was used, where data collected in the first phase were filtered using comparative tables. This phase included the translation of prime material from Japanese to the English language. Finally, a 'selective coding' method was used for the subtraction and cross-validation of data [162].

## 5. Results

### 5.1. Manufacturing

The Japanese companies were found to have significantly higher revenue and production volume than the ones in the UK. The financial, volume and machinery capacity differences are aspects highly related to their contextual conditions. In contrast, the selected companies have similar production management systems, including the selection of delay in the supply chain, manufacturing organisation, and construction systems, as seen in Table 2.

**Table 2.** Comparison of selected house manufacturing companies in Japan and the UK that were selected for this research. * Robertson Group. ** Considering Robertson Timber Engineering independent from Robertson Group.

| | Company | Revenue (M) | Volume (Houses/Year)/No. Factories | Delay Supply Strategy | Manufacturing Organisation | Structural Material | Construction System | Off-Site/On Site |
|---|---|---|---|---|---|---|---|---|
| Japan | Sekisui House | GBP 14,060 | 13,600/5 | Make to order | Flow line-like, group-like, production-line | Steel or timber | Panelised | 60/40% |
| | Sekisui Heim | GBP 7388 | 10,500/8 | Assemble to order | Flow line-like, production-line | Steel or timber | Modular | 80/20% |
| UK | Robertson | GBP 565 * | 1000/2 | Assemble to order ** | Flow line-like, workshop-like | Timber frame | Panelised | variable |
| | Scotframe | GBP 30 | 1500/2 | Assemble to order | Flow line-like, workshop-like, production-line | Timber frame | Panelised | >50/-% |
| | Carbon Dynamic | GBP 3 | <100/1 | Assemble to order | Workbench-like | Cross-laminated timber (CLT) | Modular | 85/15% |

Sekisui House highlights using a 'make to order' approach. Their system allows clients to choose structural and wall materials, including wood or steel for the structure and concrete or ceramic for walls. Each material follows a different production process. Sekisui's Heim structural material can also vary from steel to wood; however, the production line remains the same, which is why Sekisui Heim is categorised as an assembly to order. Robertson Timber Engineering, independent from Robertson Homes, works as an 'assembly to order' company as its production is delayed to the fabrication point. However, from a final customer perspective, Robertson Homes is a 'make to forecast' company as it builds houses through speculative processes.

Sekisui House and Sekisui Heim possess very high manufacturing capacities compared to Robertson, Scotframe, and Carbon Dynamic. Sekisui House is capable of producing all main construction components and parts internally from raw material, including concrete and ceramic wall panels and their structural frames, while Sekisui Heim has manufacturing lines capable of producing tri-dimensional modular units from structural wood or steel.

All companies outsource parts and components from other manufacturers, completing at least 15% of the construction on-site. Sekisui House and Sekisui Heim sub-hire local and independent contractors for the assembly of their dwellings. From a Sekisui House dwelling, only 25% of the value accounts for manufacturing and assembling in their factories; 30% is produced by suppliers of services, usually sent directly to the construction site and installed by subcontractors. Site work accounts for 20%. Sales, marketing, and management overheads account for 25%. On its part, Robertson Group distributes the supply chain into its internal departments. Robertson Timber Engineering manufactures the construction components, Robertson Construction manages the construction and assembly on-site, and Robertson Homes manage sales and land release. Robertson Timber Engineering is the only selected manufacturer not involved in the sales and design processes. Scotframe does not build houses—their clients (usually self-builders) manage the assembly of the construction

kit and other construction tasks needed. Carbon Dynamic focuses on the assembling of modules off-site and the fitting of modules on-site; other construction tasks are managed by the client, such as foundations. None of the companies cover the whole supply chain. They outsource manufacturers and contractors to cover their lack of manufacturing flexibility or capacity, and arguably to increase customisation (Figure 3).

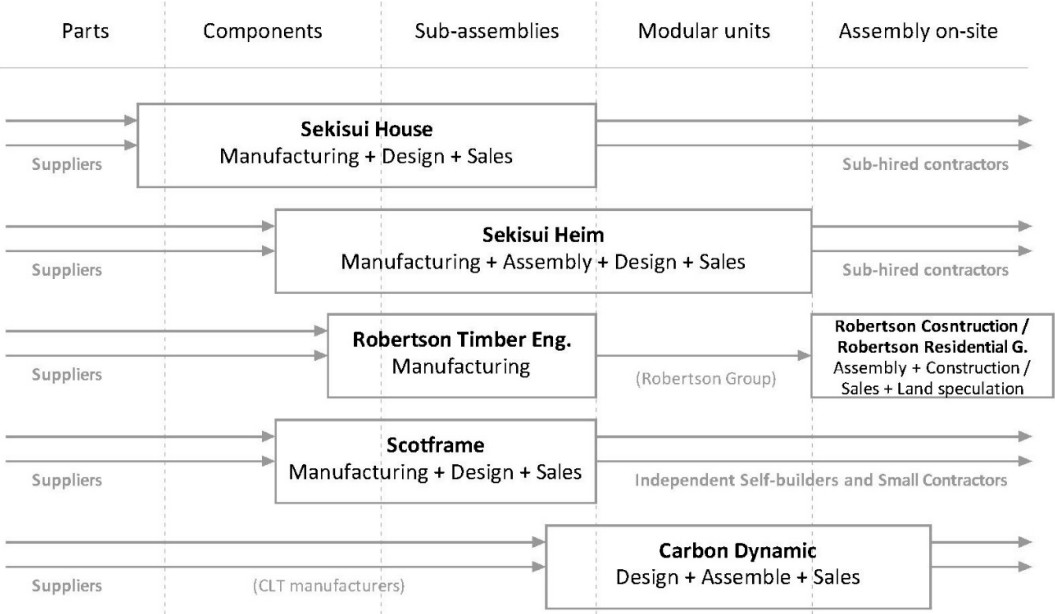

**Figure 3.** Sekisui House, Sekisui Heim, Robertson, Scotframe, and Carbon Dynamic supply chains in relation to delay strategy and outsourcing of components and services.

All companies use lean and agile manufacturing strategies; all possess systems that allow just-in-time production. Japanese manufacturers achieve variability by using efficiently organised sophisticated heavy industrial machinery. On their part, manufacturers in the UK restrict their operations to timber framing.

Sekisui Heim and Sekisui House produce all insulation materials used in their houses. However, they outsource renewables and energy efficiency mechanical systems. Daiwa House is an exception as it obtains all these components from Daiwa company; still, it is considered to be outsourcing as they are legally different companies—Daiwa House is a spin-off of Daiwa.

None of the companies in the UK produce renewables, and only Scotframe has the machinery to produce insulation material (injected to timber panels), which is optional to the client's specifications.

### 5.2. Marketing, Co-Design, and Selling Processes

Japanese companies use very sophisticated marketing strategies. They invest heavily in setting multiple highly interactive show homes/rooms around the country, putting much emphasis on informing customers about their offer. Their marketing process is not limited to informing customers about their products and services; it involves obtaining information from the customers that represent design choices, merging marketing and design into a single process—also known as a co-design process—which can be described as follows:

- Promotion—marketing strategies used to engage customers and promote the company's values. This includes brochures, home portfolios, and visitor centres, also referred to as museums. The visitor centres consist of exhibition spaces open to the public, where companies display housing-related objects and information to reinforce their unique selling point in potential consumers. For example, Daiwa's museum

focuses on showing the company's history, but also dedicates a whole floor to the exhibition of vernacular construction techniques around the world. Sekisui House focuses on informing customers about the importance of sustainable construction and energy efficiency (Figure 4).

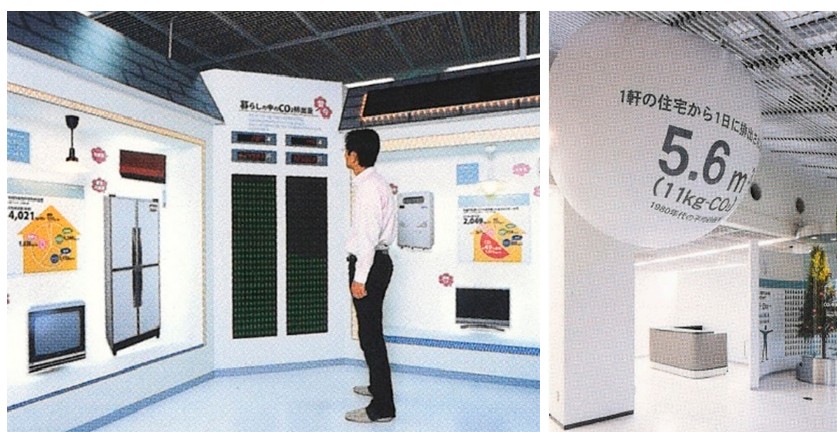

**Figure 4.** Comparison of energy consumption and carbon emissions of household appliances of the 1980s to nowadays, including solar photovoltaics (**left**) and representation of carbon emissions of a 1980s household in a balloon (**right**) at Sekisui House visitor centre.

- Show homes—facilities where customers can visit show homes and experience (see and touch) the offered house features. All Japanese companies possess various show homes around the country. Sekisui's House main housing park has an area of 18,500 m$^2$ with 21 buildings, including different show homes, prototypes, designed gardens (also on sale), screening rooms, and meeting areas. Sekisui Heim heavily populates the Japanese territory with individual show homes with design and selling agents, including show homes attached to factories (Figure 5).

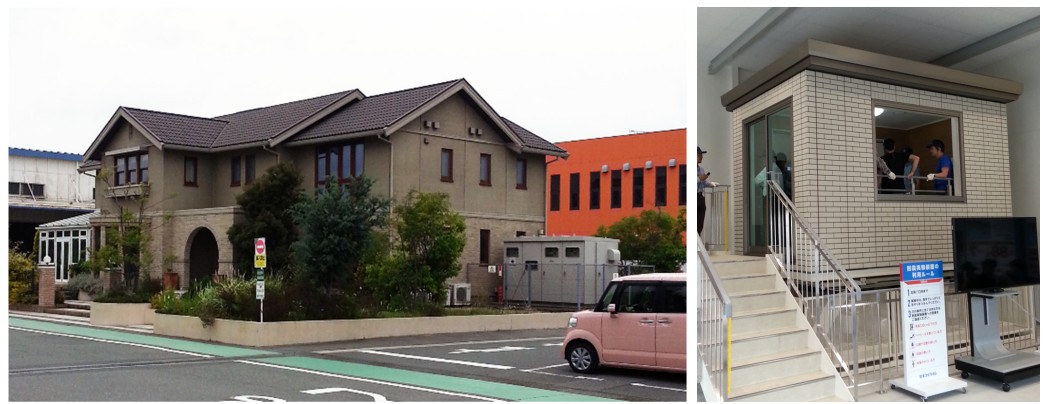

**Figure 5.** Show home (**left**) and selling point earthquake simulator (**right**) at Sekisui's Heim Aichi plant.

- Experience—strategies used to help customers make informed design decisions based on experiences. This consists of the accumulation of services provided to educate the customer on how the design decisions will affect the house performance, environmental impact, comfort, and maintenance cost. For example, Sekisui House has information centres with real-scale models of multiple house components for customers to understand their differences by interacting with them. It includes architectural features, such as lighting, staircases, and kitchen cabinets (Figure 6), as well as technical aspects related to energy efficiency and thermal performance, such as insulation, glazing, mechanical systems, and renewables (Figure 7).

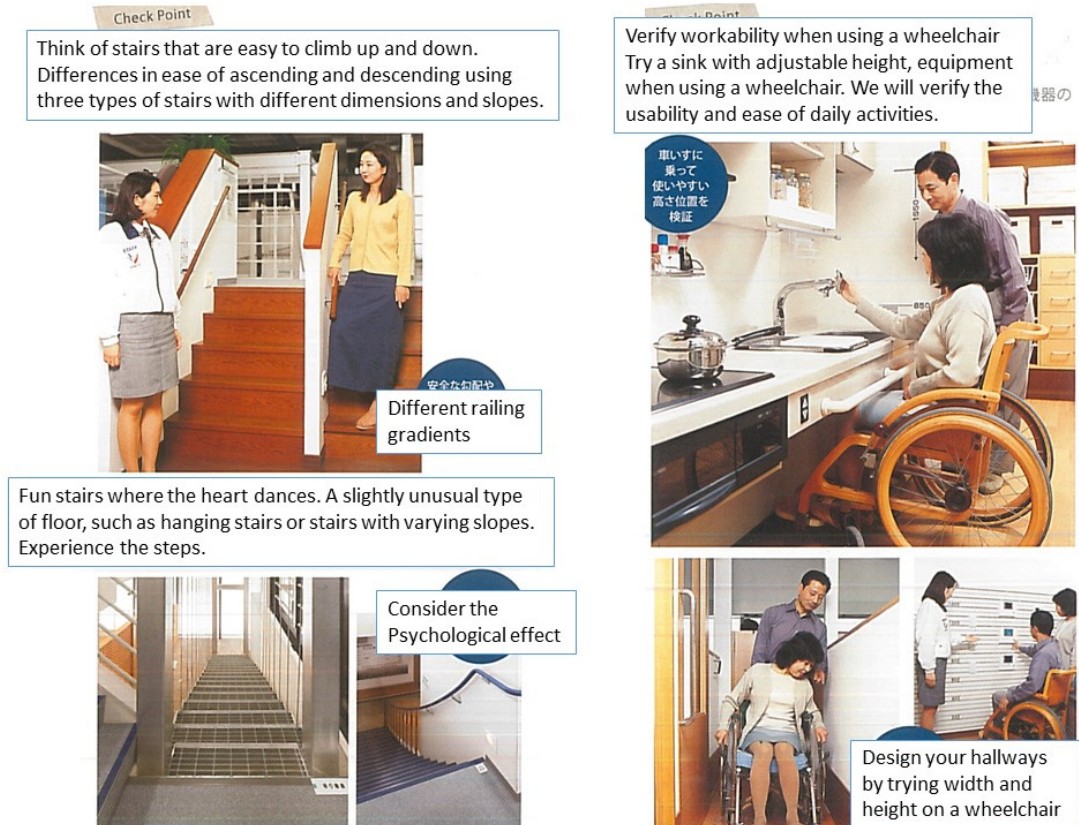

**Figure 6.** Comparison of stairs by tread width (**left**) and kitchen dimensioning with adjustable walls, cabinets, and bar heights to adapt to any type of users, including those with special needs (**right**).

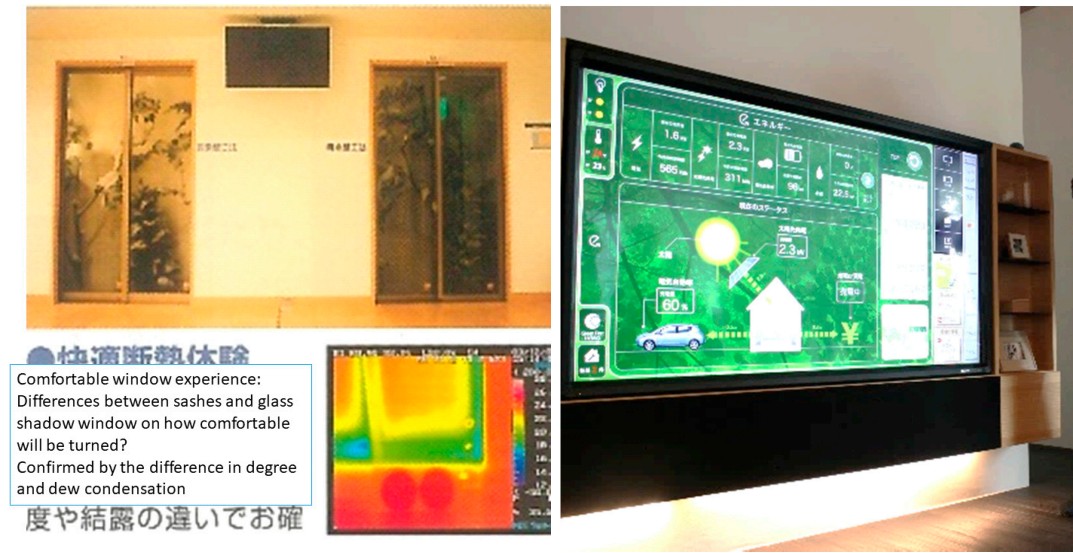

**Figure 7.** Comparison of glazing performance accompanied by thermal visualisation (**left**) and interactive home performance monitor system installed to television (**right**).

- Co-design—strategies used to extract information from the customers that represent design choices. A personal sales agent, usually an architect, is assigned to each customer for the whole designing and selling process. This process is highly linked to the 'experience' phase, as companies keep track of users' preferences to imply design decisions. The agents use responsive virtual render visualisations to seamlessly show the appearance, cost, and energy and environmental performance of the expected

dwellings. The co-design process takes place on several progressive meetings in which the customers decide some design aspects and take information home to decide on some more detailed decisions. Daiwa, for example, uses very informative brochures that include national data or testimonials from previous customers related to design-decisions, and how this could be adapted in future scenarios (Figure 8). They cover a wide range of potential design decisions with brochures for very particular living aspects, such as the selection of materials, appliances, and space arrangement depending on the customer's pet breed and size (Figure 9).

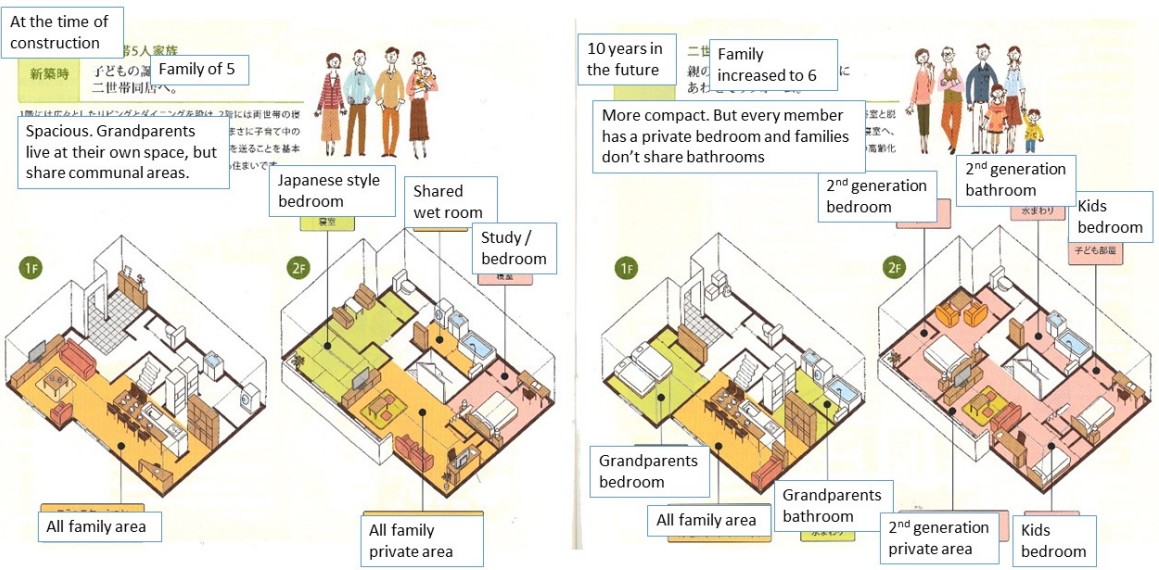

**Figure 8.** Live changing adaptation models and its corresponding architectural diagrams on Daiwa's co-living brochure.

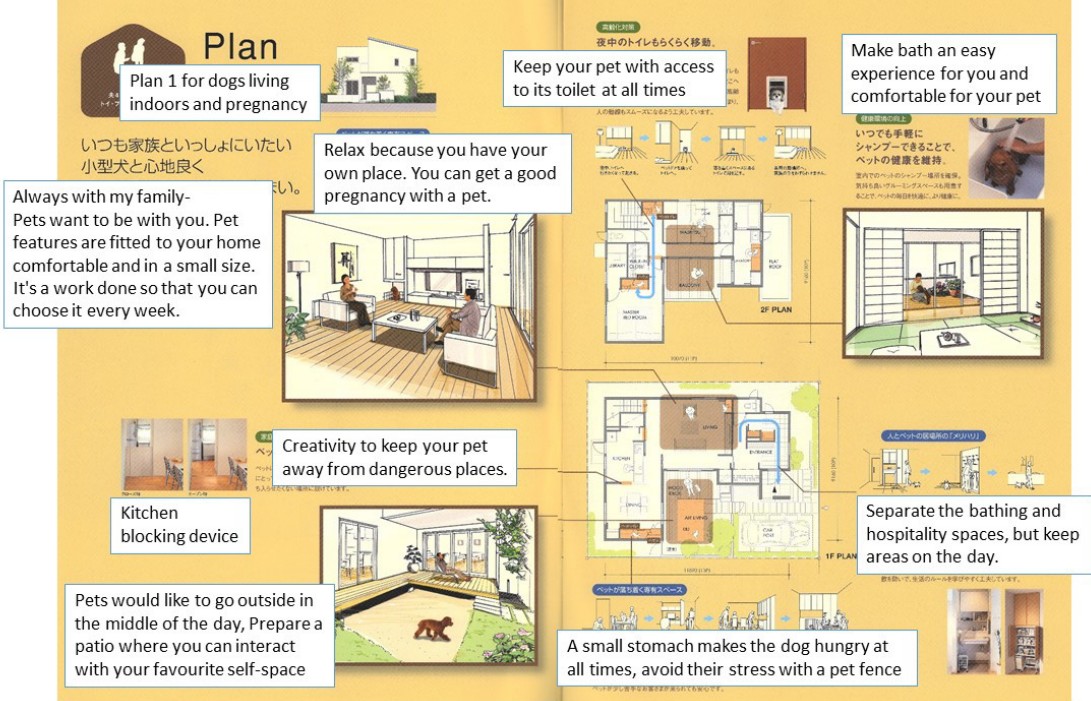

**Figure 9.** Suggested flooring materials and plan arrangements depending on for different pet breeds—Daiwa's pet brochure.

The analysed companies in the UK do have the capacity to produce mass custom houses on demand; however, they do not achieve the mass customisation levels observed in Japan as they use fewer marketing and co-design strategies that emulate pattern books or bespoke architectural design processes or lack of co-design process, as in the case of Robertson Homes (speculative sector).

For example, Scotframe has multiple home brochures that include table matrices with combinations of designed models with thermal specification and their resulting costs; however, users can only select from a list rather than being guided through a full design process (Figures 10 and 11).

**bungalows**

| BUNGALOWS House Name | Bedrooms | Floor Area M² (approx) | Bronze Open Panel | Silver Open Panel | Silver Closed Panel | Gold Closed Panel | Platinum Closed Panel | Platinum + Closed Panel |
|---|---|---|---|---|---|---|---|---|
| Acacia | 2 | 66 | £ 14,570 | £ 15,195 | £ 16,470 | £ 17,360 | £ 19,685 | £ 20,815 |
| Acer | 3 | 76 | £ 17,135 | £ 17,825 | £ 19,135 | £ 20,145 | £ 22,655 | £ 24,405 |
| Alder | 3 | 88 | £ 19,515 | £ 20,270 | £ 21,660 | £ 22,725 | £ 25,445 | £ 27,080 |
| Alder (+Garage/Utility) | 3 | 93 | £ 23,360 | £ 24,415 | £ 26,310 | £ 27,765 | £ 31,360 | £ 33,130 |
| Apple | 3 | 91 | £ 23,410 | £ 24,420 | £ 26,355 | £ 27,845 | £ 31,550 | £ 33,340 |
| Ash | 3 | 94 | £ 19,605 | £ 20,410 | £ 21,915 | £ 23,065 | £ 26,070 | £ 27,960 |
| Birch | 3 | 107 | £ 24,255 | £ 25,135 | £ 26,750 | £ 27,990 | £ 30,995 | £ 33,000 |
| Blackthorn (rear) | 3 | 112 | £ 23,425 | £ 24,300 | £ 25,890 | £ 27,130 | £ 30,150 | £ 32,470 |
| Blackthorn (front) | 3 | 114 | £ 24,820 | £ 25,730 | £ 27,355 | £ 28,655 | £ 32,070 | £ 34,685 |
| Cedar | 3 | 113 | £ 24,680 | £ 25,625 | £ 27,495 | £ 28,825 | £ 32,185 | £ 34,485 |
| Cedar (+ Garden area) | 3 | 119 | £ 28,515 | £ 29,520 | £ 31,465 | £ 32,900 | £ 36,555 | £ 39,510 |
| Cherry | 3 | 121 | £ 24,225 | £ 25,195 | £ 27,095 | £ 28,445 | £ 31,960 | £ 34,285 |
| Chestnut | 3 | 126 | £ 26,905 | £ 28,085 | £ 30,380 | £ 32,190 | £ 36,205 | £ 38,345 |
| Cypress | 4 | 127 | £ 28,790 | £ 30,090 | £ 32,635 | £ 34,455 | £ 38,995 | £ 41,355 |
| Fir (Option 1) | 3 | 135 | £ 29,725 | £ 30,800 | £ 32,755 | £ 34,300 | £ 38,060 | £ 40,970 |
| Fir (Option 2) | 4 | 141 | £ 31,665 | £ 32,770 | £ 34,775 | £ 36,365 | £ 40,260 | £ 43,420 |
| Fir (Option 3) | 4 | 145 | £ 32,065 | £ 33,205 | £ 35,260 | £ 36,880 | £ 40,865 | £ 44,020 |
| Holly | 4 | 139 | £ 30,975 | £ 32,315 | £ 34,910 | £ 36,785 | £ 41,655 | £ 44,280 |
| Larch (Split Level) | 4 | 153 | £ 32,975 | £ 34,150 | £ 36,260 | £ 37,900 | £ 42,075 | £ 44,975 |
| Magnolia | 4 | 165 | £ 37,450 | £ 39,075 | £ 42,235 | £ 44,490 | £ 50,255 | £ 53,205 |
| Mulberry (Split Level) | 4 | 171 | £ 35,985 | £ 37,245 | £ 39,635 | £ 41,380 | £ 45,815 | £ 49,155 |
| Poplar | 4 | 197 | £ 47,740 | £ 49,605 | £ 53,010 | £ 55,585 | £ 61,930 | £ 66,325 |

**Figure 10.** Bungalow design model thermal and cost matrix table on Scotframe's brochure.

| | Bronze Open Panel | Silver Open Panel | Silver Closed Panel | Gold Closed Panel | Platinum Closed Panel | Platinum Plus Closed Panel |
|---|---|---|---|---|---|---|
| **Ground Floor - by others** | | | | | | |
| Insulated Concrete Ground floor Designed / Supplied by others See assumed insulation thickness | 60mm PU | 65mm PU | 65mm PU | 120mm PU | 160mm PU | 160mm PU |
| **U' Value W/m2K (below slab)** | 0.20 | 0.19 | 0.19 | 0.13 | 0.10 | 0.10 |
| **External Walls** | | | | | | |
| Breather membrane | Reflective | Reflective | Reflective | Reflective | Reflective | Reflective |
| OSB sheathing | 9mm | 9mm | 9mm | 9mm | 9mm | 9mm |
| Framing | 140mm | 140mm | 90mm | 140mm | 235mm | 235mm |
| Insulation | 140mm Frametherm 40 | 140mm Frametherm 35 | 90mm PU | 140mm PU | 235mm PU | 235mm PU |
| OSB Sheathing | Not applicable | Not applicable | 9mm | 9mm | 9mm | 9mm |
| Vapour control layer | Not applicable | Reflective | Reflective | Reflective | Reflective | Reflective |
| Service Void / Battens | Not applicable | 35mm | 35mm | 35mm | 35mm | 35mm |
| Plasterboard | 15mm TE vapoushield | 15mm TE Plain | 15mm TE Plain | 15mm TE Plain | 15mm TE Plain | 15mm TE Plain |
| **U' Value W/m2K** | 0.25 | 0.20 | 0.20 | 0.15 | 0.10 | 0.11 |
| **Roof -Horizontal Ceiling** | | | | | | |
| Insulation | 280mm Frametherm 40 | 370mm Frametherm 40 | 370mm Frametherm 40 | 420mm Frametherm 40 | 560mm Frametherm 40 | 560mm Frametherm 40 |
| **U' Value W/m2K** | 0.15 | 0.11 | 0.11 | 0.10 | 0.08 | 0.08 |
| **Roof -Coomb/Sloping** | | | | | | |
| Insulation | 140mm PU | 170mm PU | 170mm PU | 170mm PU | 170mm PU +25mm PU | 170mm PU +25mm PU |
| Vapour control layer | Not applicable | Not applicable | Not applicable | Reflective | Reflective | Reflective |
| Service Void / Battens | Not applicable | Not applicable | Not applicable | 35mm | 35mm | 35mm |
| Plasterboard | 15mm TE vapourshield | 15mm TE vapourshield | 15mm TE vapourshield | 15mm TE Plain | 15mm TE Plain | 15mm TE Plain |
| **U' Value W/m2K -Tiled Roof** | 0.21 | 0.17 | 0.17 | 0.16 | 0.13 | 0.13 |
| **U' Value W/m2K -Slate Roof** | 0.19 | 0.16 | 0.16 | 0.14 | 0.12 | 0.12 |
| **Roof -Hanging post** | | | | | | |
| Insulation | 140mm Frametherm 40 | 140mm Frametherm 35 | 140mm Frametherm 35 | 170mm PU | 170mm +25mm PU | 170mm +25mm PU |
| Vapour control layer | Not applicable | Not applicable | Not applicable | Reflective | Reflective | Reflective |
| Service Void / Battens | Not applicable | Not applicable | Not applicable | 35mm | 35mm | 35mm |
| Plasterboard | 15mm TE vapourshield | 15mm TE vapourshield | 15mm TE vapourshield | 15mm TE Plain | 15mm TE Plain | 15mm TE Plain |
| **U' Value W/m2K** | 0.28 | 0.26 | 0.26 | 0.16 | 0.13 | 0.13 |
| **External Joinery - W/m2K** | | | | | | |
| Windows (whole product) | 1.40 | 1.40 | 1.40 | 1.40 | 1.40 | 0.90 |
| External doorsets (average product) | 1.40 | 1.40 | 1.40 | 1.40 | 1.40 | 1.40 |

**Figure 11.** Thermal kit specification supportive table on Scotframe's brochure.

The Japanese manufacturers use more marketing and strategies than the housing companies in the UK, and consequently provide more custom products (Table 3).

**Table 3.** Marketing and design strategies of selected companies.

| Marketing Strategy | | Japan | | | UK | | |
|---|---|---|---|---|---|---|---|
| | | Sekisui House | Sekisui Heim | Daiwa House | Robertson | Scotframe | Carbon Dynamic |
| Promotion | Marketing brochures | X | X | X | - | X | X |
| | Portfolio | X | X | X | X | X | X |
| | Factory visits | X | X | - | - | X | X |
| | Visitor centres 'museums' | X | - | X | - | - | - |
| Show homes | Selling points | X | X | X | X | - | - |
| | Show homes | X | X | X | X | - | - |
| | Show villas/house parks | X | X | X | - | - | - |
| | Product showroom | X | X | X | - | X | - |
| | Prototype show homes | X | - | - | - | - | - |
| Experience | Information centres | X | - | X | - | - | - |
| | Experience measurements | X | - | - | - | - | - |
| | Technology showrooms | X | X | X | - | - | - |
| Co-design | Catalogue of houses | X | X | X | - | X | X |
| | Catalogue of features | X | X | X | - | X | X |
| | Virtual interactive brochures | X | X | X | - | - | - |
| | Previous customers brochure | X | - | X | - | - | - |
| | Online configurator | - | - | - | - | - | X |
| | Assisted design | X | X | X | - | X | X |

The "experience" category is exclusive to the Japanese context, which is highly related to the company's revenue and production volume. Small and medium companies, such as Carbon Dynamic, do not have enough revenue or volume to implement these strategies.

*5.3. Energy Efficiency Offer*

From the analysed companies, only the Japanese companies offer energy-efficient mechanical systems and renewables, which are essential for conceiving zero energy/carbon houses. They include these features as customisable options rather than standards. For example, all Sekisui's House dwellings include photovoltaic solar panels (PVs), but these can be customised in type, style, size, and capacity. They offer PVs shaped as traditional ceramic tiles, which are not as efficient as conventional PVs but are appealing to some customers. Sekisui House also offers equipment related to renewables as additional options, such as different types of batteries, power cells, electric car chargers, and energy performance monitoring systems. These features are displayed in their showrooms and information centres and explained in detail in their brochures and are integrated to the co-design and selling processes for users to visualise their cost, performance, and carbon emissions. The following table shows the environmental features offered by the companies analysed in this study (Table 4).

**Table 4.** Sustainable features offered as a choice by the selected companies. * Regulations in the UK force house providers to offer a 10-year warranty, which is usually provided by external suppliers.

| Energy-Efficient/Zero Energy Equipment or Service | | Japan | | | UK | | |
|---|---|---|---|---|---|---|---|
| | | Sekisui House | Sekisui Heim | Daiwa House | Robertson | Scotframe | Carbon Dynamic |
| Fabric | Structural material (steel or wood) | X | X | - | - | - | - |
| | Insulation level | X | X | X | - | X | - |
| | | X | X | X | - | X | - |
| | Window U-value | X | X | X | - | - | - |
| | Doors U-value | X | X | X | - | - | - |
| Mechanical systems | Heating systems | X | X | X | - | - | - |
| | Ventilation systems | X | X | X | - | - | - |
| | Monitoring systems | X | X | X | - | - | - |
| | Energy cells | X | X | - | - | - | - |
| | Heat pumps | X | X | X | - | - | - |
| | Electric car charger | X | X | X | - | - | - |
| Renewables | Photovoltaics | X | X | X | - | - | - |
| Passive strategies | Green curtain | X | - | - | - | - | - |
| | Water recycling systems | X | X | - | - | - | - |
| | Solar water heater | X | X | X | - | - | - |
| Customer service | Warranty | X | X | X | * | * | * |
| | Maintenance | X | X | X | - | - | - |
| | Rearrangement | X | X | - | - | - | - |
| | Re-customisation | - | X | - | - | - | - |

The offer of sustainable features by companies in Japan is vast, while in the UK is very limited. Only Carbon Dynamic installs PVs on demand; however, these are not offered as options, and neither have they mediums to inform their customers about its environmental impact.

Housing companies in the UK do not see an advantage in including mechanical systems and renewables as they focus on profiting only from the products they manufacture. As an example, Scotframe does not include renewables because there is no apparent market demand for them, and they do not add any value. A Scotframe sales representative explains this as follows,

> *'Most clients don't really want renewables. . . . They don't also like the fact that they need to maintain them. . . . thermal heat pumps . . . it's far too complicated for them. . . . Moreover, renewables are imported, so doesn't help the economy'.*

Japanese companies subsidise a great portion of the dwelling components to add value to their houses, particularly regarding mechanical systems and renewables. Sekisui House and Sekisui Heim outsource all of the PVs, batteries, electric vehicle chargers, and power cells. Their solution spaces, manufacturing process, supply chains, and navigation tools are designed to integrate external components as integral aspects of their offer.

## 6. Discussion

### 6.1. Relation of Mass Customisation with Zero-Energy Housing

The main difference between the Japanese and UK companies is the offer of sustainable features as design choices. Japanese companies not only offer to include these features but provide multiple variants of the same feature to adjust to the customer's budget, environmental aim, and aesthetic desires.

Co-design, marketing, and selling processes have proved to be an integral aspect of mass customisation. Japanese companies promote environmental features as equal as the features and materials produced in their facilities, as they increase the market range while adding value to their products. Accordingly, they design their solution space and

navigation tools (brochures, show homes, experiences, virtual rendering, etc.) to include environmental features as a core part of their products and services. All Japanese companies provide maintenance and replacement of equipment, renewables, and mechanical systems; understanding these services as an opportunity to expand their business and increase their market appeal.

Companies in the UK wanting to specialise in sustainable housing should integrate environmental features as part of their design solution space, and consequently as part of their housing offer. It will allow them to (1) achieve higher environmental/energy levels, such as Passivhaus, zero-carbon, or zero-energy; (2) provide accurate performance and cost specifications to encourage customers for the best practice; and (3) have the potential to expand their services and increase profit by adding value to their houses.

### 6.2. Adoption of Technology Is Not On

House manufacturers in the UK already possess the robust capacity—understood as modular production, flexible automation, and flexible workforce—to produce mass custom on-demand houses. Investing in manufacturing capacity implies costs that could risk the companies' economic stability, but more importantly would not mandatorily help them to achieve mass customisation.

Industrial production benefits from constant consumption ideally at the maximum of its capacity to comply with loan paybacks and to ensure that facilities and workforce are efficiently used, which is difficult to predict in volatile markets, such as housing. Moreover, housing costs, different to other markets such as the car or shoe industries, are regulated by the existing stock. Implementing heavy or new manufacturing technologies imply a rise in production cost. Industrialised houses cost around 15% more than the average, which would entail a shift in the companies' market scope, and thus could risk its whole business foundations. Companies such as Huf Haus in Germany and Sekisui House in Japan market themselves in the high-end market (luxury) and invest in technology to achieve the quality standards of their particular market niche.

Housing companies in the UK should only invest in manufacturing technology if they do not possess the capacity, or cannot subsidise the production of all construction elements that allow them to produce zero-energy dwellings, as long as it does not imply that the price of their houses exceeds the market niche limits.

### 6.3. The Potential of Mass Customisation in the Context of Sustainable Housebuilding in the UK Relies on Improving the Housebuilders' Solution Space and Navigation Tools

Achieving full mass customisation requires not only possessing flexible robust capacity but also the development of inclusive solution spaces and appropriate navigation tools. The effectiveness of mass customisation depends on the manufacturer's capability to understand customers' needs and reflect them in their solution space, as well as having the appropriate navigation tools to show customers their offer in a clear and simple manner.

High-maintenance marketing and co-design strategies used by the Japanese companies, such as the information centres, museums, technology showrooms, and housing parks, are above the budget of the small- and medium-housing companies in the UK, such as Carbon Dynamic. The design of the solution space and selection of navigation tools need to adapt to cultural and economic aspects of the context in which the company stands. Technology showrooms are crucial to Japanese companies because of the importance that resistance to natural disasters means for the built environment, while information centres are associated with their cultural interest in technology and amusement; however, this is not the case for buildings in the UK.

Companies in the UK need to expand their marketing and co-design strategies along with the mass customisation capabilities (robust design, solution space, and navigations tools) for these to have an impact on their sales and market focus. For example, Scotframe's solution space contemplates a combination of six levels of fabric insulation and more than 100 house models; however, they lack navigation tools that clearly explain and define the implications of each choice. Sekisui Heim, in contrast, has the capacity to produce higher

variability than Scotframe, but presents customers an apparent offer of only 22 models from which all the possible variants are gently added through the co-design process. Carbon Dynamic, for their part, possess online configurators and virtual reality platforms where users can choose and visualise the different design offers, including finishing materials and room arrangements. However, this does not present pricing and is not included as part of the design decision making process.

The companies in the UK should implement marketing strategies according to their financial capacity and market positioning. None of the selected companies in the UK have show homes, which is an effective strategy used by all Japanese companies and other house manufacturers present in the UK, such as 'Huf Haus'. Show homes might be out of the budget of small companies, such as Carbon Dynamic; however, companies that expect to implement mass customisation need to invest in marketing strategies. The implementation of brochures, for example, does not imply a significant investment. All the companies selected use brochures and catalogues to promote and show their house models. Even Robertson Homes provides catalogues of their houses in stock. However, only the Japanese companies use brochures as design guides. Daiwa's brochures are highly sophisticated and include multiple information techniques, such as graphics, diagrams, sketches, architectural plans, design examples, and narratives of previous customers. Daiwa is recognised for providing the most user-friendly selling process and the highest level of customisation, and they do it without the high-maintenance experience strategies used by Sekisui House.

Housing companies in the UK could release the mass customisation potential of their manufacturing capacity and solution spaces using low-cost marketing strategies, such as brochures, as long as they are designed as part of a navigation toolset and developed according to marketing research.

### 7. Conclusions

This paper described the relationship that mass customisation has with the production and consumption of sustainable housing through a comparative analysis between house manufacturers in Japan and UK. It also described how Japanese manufacturers are using mass customisation strategies to allow end-users to customise their houses in detail, including a diverse range of environmental features, while effectively communicating the dwelling's operational energy costs and carbon impacts with sophisticated tools, visuals, catalogues, guides, and models. Mass customisation techniques are allowing Japanese house manufacturers to provide their customers with information to make informed choices, which has consequently resulted in the lead of production of zero energy and zero carbon houses.

The current attempts to improve housing production in the UK, as modern methods of construction, are mainly focused on solving manufacturing constraints, rather than focusing on improving the service provided to house buyers—price certainty, high customisability, and a sustainable offer that covers the market wants and needs. The UK construction industry has been sceptical about implementing mass customisation because it seems to be exclusive of other markets. However, mass customisation belongs to all production practices and services, in which housing is both; its use in Japan has proved its feasibility to the housing and construction contexts.

This paper concludes that housebuilders in the UK could adopt mass customisation and customer-oriented strategies to gain an advantage in the housing market, particularly in the rising niche of sustainable housing. Energy efficiency is the most recognisable sustainability tag and has proved to be a feature that house buyers are looking for in a house; therefore, it is in the interest of housebuilders to produce energy-efficient houses (zero-energy, zero-carbon, Passivhaus, etc.) to distinguish themselves in the market and succeed as businesses. Therefore, mass customisation can be seen as one of the multiple paths towards sustainability in the UK housing context that has struggled to find innovative methods to supply sustainable housing.

Housebuilders interested in adopting mass customisation for the delivery of sustainable housing should focus on developing solutions spaces that include a wide range of environmental features and in sophisticating their co-design, marketing, and communication strategies to use them as design navigation tools based on appropriate market research.

**Funding:** This research was funded by the Mexican National Council of Science and Technology (CONACYT).

**Institutional Review Board Statement:** Not applicable.

**Informed Consent Statement:** Not applicable.

**Data Availability Statement:** Data are available in a publicly accessible repository.

**Acknowledgments:** The author would like to acknowledge the guidance of John Brennan; support of Masa Noguchi, Norrie Smith, and the ZEMCH Network; and the help of Johanna Morey.

**Conflicts of Interest:** The author declares no conflict of interest. The funders had no role in the design of the study; in the collection, analyses, or interpretation of data; in the writing of the manuscript; or in the decision to publish the results.

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
