# Peer review of "Mass Customisation for Zero-Energy Housing"

_sustainability, doi:10.3390/su13105616_

Round 1
Reviewer 1 Report
Dear Author,
your manuscript seems to be very interesting. I appreciate your work. But there are some minor issues to be revised in my opinion.
Resoultion of figures is to low, and the figures are hard to read.
The chapter 2 is very long and the content is unevenly didtributed. Subsection 2.2. is very short compared to others. It contains one paragraph only. I suggest to extend it by the analysis of some sources relating to mass customisation in housebuilding components industry including the articles published in this journal prevoiusly f.e. su12093788.
It should be explained in the main text why Daiwa Homes was excluded from the large part of the analysis. Why didn't you find an other more familiar company from Japan?
Author Response
Find attached the article with the corrections suggested. I have included statements in the introduction and conclusions that explain how the article relates to the topic of the special issue. Chapter number 2 has been briefed, while section 2.2 has been extended. Images have been uploaded with better quality, on a readable scale.
Kind Regards,
Pablo

Reviewer 2 Report
This is a well written, well presented and well argued article. The author shows a deep knowledge and understanding of the subject. The methods of this research are clearly explained and connects well with the presentation and discussion of results.
The article is very strong. It offer a great learning opportunity between Japan and the UK, which contributes to practice. In this the UK housing industry can learn a great deal from the Japanese context. However, there is a social, political, cultural, economic and environmental differences between the two countries, which needs to be understood within local realities. Something for the author to think about in future research and publications.
Overall, I do not have any comments other than congratulating on this article.
Author Response
Thank you for your useful feedback. I agree that understanding a delimiting the cultural and social barriers between the UK and Japan are highly important and something worth researching forward; otherwise implementing their ways could be counterproductive.
I will consider this factor in my further research
Pablo